# Body Shape Phenotypes and Breast Cancer Risk: A Mendelian Randomization Analysis

**DOI:** 10.3390/cancers15041296

**Published:** 2023-02-17

**Authors:** Laia Peruchet-Noray, Niki Dimou, Anja M. Sedlmeier, Béatrice Fervers, Isabelle Romieu, Vivian Viallon, Pietro Ferrari, Marc J. Gunter, Robert Carreras-Torres, Heinz Freisling

**Affiliations:** 1International Agency for Research on Cancer (IARC/WHO), Nutrition and Metabolism Branch, CEDEX 08, 69372 Lyon, France; 2Department of Clinical Sciences, Faculty of Medicine, University of Barcelona, 08007 Barcelona, Spain; 3Department of Epidemiology and Preventive Medicine, University of Regensburg, Franz-Josef-Strauss-Allee 11, 93053 Regensburg, Germany; 4Département Prévention Cancer Environnement, Centre Léon Bérard, CEDEX 08, 69373 Lyon, France; 5National Institute of Public Health, Cuernavaca 62100, Morelos, Mexico; 6Digestive Diseases and Microbiota Group, Institut d’Investigació Biomèdica de Girona Dr. Josep Trueta (IDIBGI), 17190 Salt, Spain

**Keywords:** body shape, breast cancer, height, Mendelian randomization, obesity

## Abstract

**Simple Summary:**

Investigating body shapes may offer novel insights into the role of adiposity and body size in breast cancer development. We derived three distinct body shapes from combinations of height, weight, body mass index (BMI), waist and hip circumference, and waist-to-hip ratio (WHR) by a principal component analysis. We then used genetic variants linked to these body shapes and investigated associations with breast cancer risk. A body shape characterizing general adiposity was associated with a lower breast cancer risk, however this is likely because the genetic variants predicting this body shape reflect body fatness during childhood and adolescence rather than during later adulthood. A body shape characterizing tall women with a low WHR was weakly associated with a higher breast cancer risk. A body shape characterizing tall women with a large WHR was not associated with breast cancer risk. These findings can spur wider application in cancer research and possibly risk stratification.

**Abstract:**

Observational and genetic studies have linked different anthropometric traits to breast cancer (BC) risk, with inconsistent results. We aimed to investigate the association between body shape defined by a principal component (PC) analysis of anthropometric traits (body mass index [BMI], height, weight, waist-to-hip ratio [WHR], and waist and hip circumference) and overall BC risk and by tumor sub-type (luminal A, luminal B, HER2+, triple negative, and luminal B/HER2 negative). We performed two-sample Mendelian randomization analyses to assess the association between 188 genetic variants robustly linked to the first three PCs and BC (133,384 cases/113,789 controls from the Breast Cancer Association Consortium (BCAC)). PC1 (general adiposity) was inversely associated with overall BC risk (0.89 per 1 SD [95% CI: 0.81–0.98]; *p*-value = 0.016). PC2 (tall women with low WHR) was weakly positively associated with overall BC risk (1.05 [95% CI: 0.98–1.12]; *p*-value = 0.135), but with a confidence interval including the null. PC3 (tall women with large WHR) was not associated with overall BC risk. Some of these associations differed by BC sub-types. For instance, PC2 was positively associated with a risk of luminal A BC sub-type (1.09 [95% CI: 1.01–1.18]; *p*-value = 0.02). To clarify the inverse association of PC1 with breast cancer risk, future studies should examine independent risk associations of this body shape during childhood/adolescence and adulthood.

## 1. Introduction

Breast cancer is the most frequently diagnosed cancer in women worldwide (2.26 million new cases in 2020) and is the leading cause of cancer mortality in women (684,996 deaths in 2020) [1]. Mammography screening reduces breast cancer mortality effectively by detecting breast cancer at an early and curable stage [2]. In addition, mutations in high penetrance genes (e.g., BRCA1/2) can help to identify individuals at a higher risk of developing breast cancer [3]. However, the breast cancer screening participation rate varies largely across countries and can be as low as 16% [4] and the known genetic predisposing genes account for 5–10% of the total breast cancer burden [5]. The identification of breast cancer risk factors is crucial to reduce incidence and associated mortality, and to risk-stratify the population to optimize screening participation [2].

Established risk factors for female breast cancer include age, family history, alcohol consumption, ionizing radiation exposure, reproductive factors, such as early menarche, late menopause, late age at first pregnancy, low parity, and elevated levels of both endogenous and exogenous estrogen [6].

There is inconsistent evidence from prospective observational and genetic studies linking body fatness, mostly assessed by body mass index (BMI), to breast cancer incidence [6,7]. Mostly from observational studies, BMI has been reported to be either positively or inversely associated with breast cancer depending on menopausal status and the molecular breast cancer sub-type [6]. However, genetically predicted BMI is inversely associated with breast cancer risk, independently of the individual’s menopausal status [7]. Additionally, other single anthropometric traits have also been assessed as risk factors for breast cancer risk. For instance, waist circumference (WC) has been positively associated with breast cancer risk among post-menopausal women, while the association with waist-to-hip ratio (WHR) remains inconclusive [6]. Finally, height has been positively associated with a risk of overall and hormone-receptor-positive breast cancer [6,8].

While a correlation between different anthropometric variables exists, individuals with similar values in one trait (e.g., BMI) may differ considerably for others (e.g., WHR) resulting in different body shapes. A promising approach to capture the complexity of body shape phenotypes is to combine multiple traits. In 2016, a principal component analysis (PCA) was performed on six anthropometric traits (height, weight, BMI, WC, hip circumference (HC), and WHR) to define body shape [9]. Each principal component (PC) represented a specific pattern of the six anthropometric traits and captured distinct body shape dimensions. Different biological mechanisms may underlie the heterogeneity of body morphology with breast cancer risk, and the different body shapes may confer differences in breast cancer risk. We have previously shown that these body shapes were differentially associated with breast cancer risk in an observational analysis [10]. For example, a body shape characterizing overall adiposity was positively associated with post-menopausal breast cancer risk, but not associated with pre-menopausal breast cancer risk, while a body shape contrasting tall women with large WHR and short women with low WHR was not associated with either pre- or post-menopausal breast cancer [10]. However, considerable uncertainty remains about the potential causal effect of these anthropometric phenotypes, individually or in combination, on overall breast cancer risk and according to sub-type.

Mendelian randomization (MR) analysis uses genetic variants as instrumental variables (IVs) to strengthen the causal inference in exposure–outcome associations in the absence of pleiotropic effects (i.e., a single genetic locus influences multiple phenotypes). This methodology can overcome some of the limitations of conventional approaches, such as reverse causation and unmeasured confounding, provided that the key assumptions of MR are not violated [11].

In this study, a two-sample MR analysis was conducted to investigate the potential causal associations between three body shape phenotypes, as derived in Ried et al. [9], and the risk of breast cancer using summary level data including 133,384 breast cancer cases and 113,789 controls from the Breast Cancer Association Consortium (BCAC) [12]. We also investigated associations by hormonal receptor status breast cancer sub-types.

## 2. Materials and Methods

### 2.1. Body Shape Phenotypes and Related Genetic Variants

Ried et al. performed a PCA of six anthropometric traits (BMI, height, weight, WC, HC, and WHR) obtaining three body shape phenotypes [9]. Each PC represents a specific composition of the six anthropometric traits and thus captures a specific aspect of body shape. The first three PCs explained 96.7% of the variation of these six anthropometric traits [9]. PC1 explained on average 64.4% of the variation in all traits and had equally high positive loadings on all traits, except for height, characterizing individuals with general adiposity. PC2 explained 18.5% of the variation and had high but opposite loadings on height and WHR, capturing variation in a composite body shape that represents tall individuals with a low WHR or vice versa, short individuals with a large WHR. PC3 explained 13.8% of the variation and had high positive loadings on height and WHR, and an opposite loading of nearly the same size on HC. PC3 characterizes tall individuals with a large WHR resulting from a smaller HC or vice versa, short individuals with a low WHR and a high HC [9]. Further details regarding PC loadings can be found in Appendix A.

Ried et al. also reported a fourth body shape (PC4) characterizing individuals with high BMI and weight with low WC and HC (athletic individuals) and vice versa [9]. Although we also attempted to analyze PC4 as an additional exposure, the results obtained are not reported in this manuscript due to their imprecision and the impossibility of building solid conclusions.

The genome-wide association study (GWAS) to identify genetic variants robustly associated with the body shape phenotypes included more than 170,000 individuals of European descent from 65 studies from the GIANT Consortium. All study participants gave written informed consent and ethic committees approved all studies. The results included significant associations (*p* < 5 × 10^−8^) of 200 loci (31 for PC1, 124 for PC2, and 45 for PC3), some of them not previously associated with single anthropometric traits [9]. On this basis, we excluded 11 genetic variants after linkage disequilibrium analysis (LD r^2^ ≤ 0.01) to avoid correlation between genetic variants, using the 1000 Genomes Project European samples. Genetic variant summary statistics for associations with PC1–3 were obtained for the remaining 189 genetic variants (30 for PC1, 115 for PC2, and 44 for PC3, considering 4 genetic variants related to both PC2 and PC3) Appendix A.

### 2.2. Breast Cancer Risk Data

Genetic variant summary statistics for associations with the risk of overall breast cancer and 5 sub-types were extracted from a GWAS of 247,173 individuals (133,384 breast cancer cases and 113,789 controls) of European ancestry carried out by the Breast Cancer Association Consortium (BCAC) [12]. The overall breast cancer analyses were based on mostly middle-aged women of European ancestry from 82 BCAC studies from over 20 countries including controls, cases of invasive breast cancer, cases of carcinoma in situ and cases of unknown invasiveness. In the sub-type analyses, 81 BCAC studies were included, restricted to controls and cases of invasive breast cancer. The study comprised data on luminal A (7325 cases), luminal B (1682 cases), HER2+ (718 cases), triple negative (2006 cases) and luminal B/HER2 (1779 cases) invasive breast cancer sub-types, and 20,815 controls. More detailed information on sample size and tumor heterogeneity is provided in Appendix A. All participating studies were approved by their appropriate ethics or institutional review board and all participants provided informed consent.

For 6 genetic variants, the use of a proxy genetic variant was needed (r^2^ ≥ 0.8; European samples), while for 1 genetic variant, no data were obtained.

### 2.3. Statistical Analysis

We performed a two-sample MR analysis to estimate the causal relation between independent body shape phenotypes (PC1, PC2 and PC3) and breast cancer risk (Figure 1).

The MR analysis was performed and reported in line with the STROBE-MR guidelines [13,14]. The STROBE-MR guidelines checklist can be found in Appendix A. MR is based on three key assumptions: (1) the genetic variants are associated with the exposure, (2) the genetic variants are not associated with any potentially confounding variable, and (3) the genetic variants are only associated with the outcome through the risk factor [15,16]. To validate the first MR assumption, we used genetic variants identified from a two-stage meta-analysis of GWAS performed on the three first body shapes [9].

We used the fixed-effects inverse-variance weighted (IVW) method as the primary approach [17]. This analysis can be understood as a meta-analysis of the Wald estimates for each genetic variant effects which are calculated as the ratio of the genetic variant-breast cancer over the genetic variant-body shape association [15,18]. The fixed-effects IVW model assumes absence of horizontal pleiotropy and will return an unbiased estimate provided that all variants are valid instrumental variables. We calculated the MR-derived odds ratio (OR) of breast cancer risk for a one standard deviation (SD) increase in genetically predicted body shape phenotype.

### 2.4. Sensitivity Analyses

As the second (i.e., genetic variants are not associated with any potentially confounding variable) and the third (i.e., genetic variants are only associated with the outcome through the risk factor) MR assumptions are challenging to test [19], further analyses were performed to assess any potential violation of both assumptions.

First, we used PhenoScanner (available at http://www.phenoscanner.medschl.cam.ac.uk, accessed on 12 June 2021) and the GWAS catalog (available at https://www.ebi.ac.uk/gwas/, accessed on 15 June 2021) to investigate whether our selected genetic variants were associated with other traits in previous GWAS [19,20,21].

Second, we used Cochran’s Q test to evaluate the heterogeneity of causal effects for each variant, with a *p*-value < 0.05 indicating statistical significance [22]. In the case of heterogeneity, likely indicating pleiotropy [23], a random-effects IVW MR analysis was used [24]. This method relaxes the third assumption as the total pleiotropic effect of a single genetic variant no longer needs to be null but assumes a zero mean between all the genetic variants (i.e., balanced pleiotropy) [24]. In addition, we implemented MR-Egger regression [25] and the weighted median approach [26]. The MR-Egger methodology provides an unbiased causal effect estimate even if the third MR assumption is violated and all the variants are invalid instruments. However, the Instrument Strength Independent of Direct Effect (InSIDE) assumption needs to be held. This additional assumption is based on the independency between the horizontal pleiotropic effects and the variants-exposure effects. MR-Egger also provides a statistical test for overall directional pleiotropy based on whether the intercept term is different from zero, as well as the I^2^_GX_ statistic, which indicates the potential violation of the NO Measurement Error (NOME) assumption and suggests the unreliability of MR-Egger inferences at values below 90% [25,27]. The weighted median estimator allows for violations of the second and the third assumptions when up to 50% of the genetic variants are invalid (i.e., violation of one or more of the three basic MR assumptions) [23,26]. Both tests provide valid MR estimates in the presence of overall directional pleiotropy but suffer from reduced power [25,26]. We also used the MR Pleiotropy RESidual Sum and Outlier (MR-PRESSO) method to identify and remove any outlying variants, applying a random-effects IVW model [28]. This method regresses genetic variant-outcome on genetic variant-exposure and uses the square of residuals to identify outliers. Finally, leave-one-genetic-variant-out analyses were used to assess whether any association was driven by specific genetic variants.

A two-sided *p*-value < 0.05 was considered statistically significant in all analyses.

We used scatter plots to present the genetic associations between body shape phenotypes and breast cancer risk, in combination with funnel plots, to visually examine the consistency of MR estimates and the potential associated bias (Appendix A).

All analyses were conducted with the statistical software R 4.0.4 and RStudio 1.4.1106 using the MendelianRandomization 0.6.0 and MRPRESSO 1.0 R packages [28,29].

## 3. Results

### 3.1. Body Shape Phenotypes and Risk of Overall Breast Cancer

PC1 was inversely associated with breast cancer risk (IVW_random-effects_ OR per 1 SD equal to 0.89 [95% CI: 0.81–0.98]; *p*-value = 0.016) (Figure 2). There was evidence of heterogeneity across genetic variants (*p*-value < 0.001) and of overall directional pleiotropy (Egger intercept = 0.029 [95% CI: 0.014–0.043]; *p*-value < 0.001) (Appendix A). Nevertheless, the results of the random IVW after performing MR PRESSO, MR Egger, and the weighted median supported the inverse association (Egger OR = 0.59 [95% CI: 0.48–0.74]; *p*-value < 0.001; weighted median OR = 0.81 [95% CI: 0.76–0.88]; *p*-value < 0.001) (Appendix A). The leave-one-genetic-variant-out analysis showed an estimate closer to the null when excluding the genetic variant rs1121980 (a variant in the FTO gene) (Appendix A). This genetic variant along with the rs1582874 variant (associated with height and other traits) (Appendix A) were identified as outliers by the MR-PRESSO method. No distortion of the overall risk estimate by the presence of these two outlier genetic variants was detected (ρ distortion = 0.13) (Appendix A).

PC2 was weakly positively associated with the risk of breast cancer (IVW_random-effects_ OR = 1.05 [95% CI: 0.98–1.12]; *p*-value = 0.135) (Figure 2), but the confidence interval included the null. We found the presence of heterogeneity across genetic variants (*p*-value < 0.001), but directional pleiotropy was not detected (Appendix A). Accordingly, the dispersion of the estimates provided by the genetic variants in the funnel plot showed balanced pleiotropy (Appendix A). However, MR PRESSO indicated that the presence of five outlier genetic variants was potentially biasing the initial risk estimate (ρ distortion = 0.013) (Appendix A). The weighted median and IVW random MR approaches without outlier genetic variants estimations were attenuated toward the null (Appendix A).

There was no evidence of an association between PC3 and breast cancer risk (IVW_random-effects_ OR = 1.04 [95% CI: 0.89–1.21]; *p*-value = 0.619) (Figure 2). We observed evidence of heterogeneity (*p*-value < 0.001), but without the presence of overall directional pleiotropy (Appendix A). The sensitivity analysis supported a null association (Appendix A).

### 3.2. Body Shape Phenotypes and Risk of Breast Cancer Sub-Types

PC1 was inversely associated with a risk of all five breast cancer sub-types, of which associations were most consistent for HER2+ (IVW_random-effects_ OR = 0.81 [95% CI: 0.67–0.99]; *p*-value = 0.043) and triple negative sub-types (IVW_random-effects_ OR = 0.82 [95% CI: 0.73–0.91]; *p*-value < 0.001) (Figure 2). Nevertheless, all of the associations showed heterogeneity and overall directional pleiotropy (Appendix A). Sensitivity analyses accounting for unbalanced pleiotropy and outliers supported the inverse associations of the main analysis (Appendix A). MR-PRESSO classified genetic variants as outliers in all the sub-types but only in luminal A sub-type identified biased association estimates (ρ distortion = 0.02) (Appendix A).Comparable to the overall breast cancer risk analysis, PC2 risk estimates included the null, except for luminal A (IVW_random-effects_ OR = 1.09 [95% CI: 1.01–1.18]; *p*-value = 0.02) (Figure 2) with the presence of balanced pleiotropy (heterogeneity *p*-value luminal A < 0.001; Egger intercept *p*-value luminal A = 0.256) (Appendix A).

PC3 was suggestively inversely associated with the risk of the HER2+ sub-type breast cancer (IVW_random-effects_ OR = 0.75 [95% CI: 0.53–1.08]; *p*-value = 0.122) (Figure 2), in contrast with other sub-types; triple negative and luminal B/HER2 negative sub-types of breast cancer, although risk estimates were imprecise (OR_triple negative_ = 1.14 [95% CI 0.88–1.48]; OR_luminal B/HER2 negative_ = 1.13 [95% CI 0.88–1.44]) (Figure 2). The sensitivity analysis showed comparable results except for the Egger test which presented unreliable results (I^2^_GX_~0%) similar to the overall breast cancer analysis (Appendix A).

## 4. Discussion

We performed a large-scale MR analysis to estimate potential causal associations between three distinct body shape phenotypes and the risk of overall breast cancer and five breast cancer tumor sub-types. A body shape capturing general adiposity (PC1) was inversely associated with the risk of overall breast cancer. The inverse association was stronger for ER− subtypes as compared to ER+ sub-types. A body shape contrasting tall women with a low WHR to short women with a large WHR (PC2) was weakly positively associated with overall breast cancer risk and most sub-types. There was no evidence of an association between PC3, a body shape characterizing tall women with a large WHR, and the risk of overall breast cancer. There was suggestive evidence that PC3 was differentially associated with breast cancer sub-types (inversely associated with HER2+, but positively associated with triple negative and luminal B/HER2 negative).

General adiposity has been traditionally assessed using BMI, and in the absence of MR studies using PC1, BMI is an indicator that is closest to the PC1 body shape in our study. Previous MR analyses showed an inverse association between BMI and pre- and post-menopausal breast cancer risk [7,30,31,32,33], which is congruent with our findings for PC1. Specifically, a 2019 MR analysis found that BMI was inversely associated with the risk of overall breast cancer, and ER+ (luminal A, luminal B and luminal B/HER2 negative) and ER− (HER2+ and triple negative) sub-types, using a similar dataset—although with less cancer cases—to the BCAC study [31]. However, the associations observed for the ER+ and ER− sub-groups slightly differed from our results. While our results suggest a 10% and 20% reduced risk, in ER+ and ER− related sub-types of breast cancer, respectively, Ooi et al. did not find any difference between ER+/− and reported an OR close to 0.8 for both sub-types [31].

This difference could be due to the specificity of PC1, which captures a body shape that goes beyond BMI. PC1 could thus be complementary to, and potentially more specific than, BMI in characterizing general adiposity. This assertion is supported by the finding that two novel loci for PC1 were identified by Ried et al. that were not identified before in larger GWAS analyses for BMI, WHRadjBMI, and height [9]. In comparison to other research in the field using anthropometric traits, such as BMI, waist circumference, or height individually, we captured body size and adiposity holistically by combining information from six anthropometric traits. The derived body shapes represent anthropometric information that is by construction orthogonal, i.e., mutually adjusted for the six traits and risk associations (here: with breast cancer) are likely less biased due to residual confounding between traits in standard multivariable regression e.g., adjusting WHR for height.

In contrast to the MR results, observational studies have shown that a high BMI throughout adulthood is convincingly associated with a higher risk of post-menopausal breast cancer [6,34]. Observational studies also showed that a high BMI prior to menopause is associated with a lower risk of pre-menopausal breast cancer [6]. Furthermore, a high BMI in young adulthood (18–30 years of age) is associated with a lower risk of pre- and post-menopausal breast cancer [6]. A 2020 MR study provided some potential insights into the apparently divergent findings between observational and MR approaches regarding post-menopausal breast cancer risk [35]. This MR study found that elevated childhood BMI (age 10 years) was associated with a decreased risk of pre- and post-menopausal breast cancer combined, whereas adult BMI (mean age 56.5 years) was suggestive of a positive association (OR = 1.08 [95% CI: 0.93–1.27]) after accounting for childhood BMI [35]. Body fatness in childhood and adolescence (indicated by BMI) has also been linked to a decreased risk of breast cancer among pre- and post-menopausal women in observational studies [36,37,38], which suggests a long-term effect of body fatness at a young age on breast cancer risk later in life. Mechanistic evidence from in vivo experiments in female rats supports this notion whereby adipose-tissue-derived estrogen in overweight adolescents may induce early breast differentiation and render the breast tissue less susceptible to tumorigenesis [39]. Thus, the genetically predicted PC1 exposure used in our MR analysis may reflect body fatness during childhood and adolescence rather than during later adulthood explaining the inverse association with breast cancer risk.

In post-menopausal women, adipose tissue converts adrenal androgens into estrogen by the aromatase enzyme [40]. The presence of estrogen, alone and with progestin, is directly associated with an increased post-menopausal breast cancer risk [40]. Furthermore, in vivo and in vitro models have shown that macrophages that are stimulated by an obese milieu can increase estrogen signaling in breast cancer cells through the upregulation of ERα expression [41]. Indirectly, higher levels of estrogen may upregulate leptin production, which may also be relevant for breast cancer development [40]. For example, leptin mediates immune suppression facilitating cancer formation and progression [40]. However, in pre-menopausal women, high vs low BMI is generally associated with longer anovulatory cycles resulting in lower levels of estrogen and progestin [42], thereby potentially providing a barrier to cancer formation. Taken together, the time of onset of excess body fatness and its modulating effects on circulating levels of estrogen and progestin may partly explain the divergent associations of body fatness with breast cancer risk. Although we were not able to differentiate between pre- and post-menopausal breast cancer with our summary-level genetic data, our results remain consistent with other MR analyses that used individual-level data and found that BMI was inversely associated with breast cancer risk, independently of menopausal stage [30,33].

Our findings concerning PC2 (tall with low WHR) are in line with the observational analysis of a previous publication of our group, where PC2 was positively associated with pre- and post-menopausal breast cancer with risk increments of 8–10% per 1 SD increase [10]. Positive associations between height and breast cancer risk using both observational and instrumental approaches have been reported with risk increases of 7–20% per ~10 cm increase [8,43]. Our observational and MR results for PC2 (OR = 1.05 [95% CI: 0.98–0.13]) were slightly weaker than those reported in the literature for height, which can potentially be explained by the contribution of WHR into PC2. However, no clear role of WHR with cancer risk has been observed [32,44].

Regarding breast cancer sub-types, only PC2 showed a positive association with the luminal A (ER+/PR+) tumor subtype. The results of other sub-types showed a confidence interval including the null. Zhang et al. reported a positive association between height and all hormone receptor-positive (ER+/PR+) sub-type breast cancers and no relationship for hormone receptor-negative (ER−/PR−) sub-types [8]. The lack of a consistent positive association between PC2 and other hormone receptor-positive sub-types (i.e., luminal B and luminal B/HER2 negative) in our analysis could be due to the lower number of genetic variants linked to height (114 variants; see Appendix A) when compared to Zhang et al. (168 variants) [8].

Mechanistically, breast tissue in taller women is thought to be exposed to higher levels of insulin, growth hormone and IGF-I, and thus may have undergone more cell divisions contributing to a greater potential for error during DNA replication [6]. Animal models clearly support a key role of growth hormone and its downstream signaling, including the IGF pathway, in breast cancer development [45].

For body shape PC3, indicating body height combined with small HC and large WHR, we found little evidence for an association with breast cancer risk and sub-types, congruent with observational results from our group [10]. However, HER2+ sub-type showed an inverse association trend, in contrast with other breast cancer sub-types. HER2+ sub-type showed inverse associations for all tested PCs, being only statistically significant for PC1. In the previous literature, HER2+ sub-type tumors have been identified as a distinct subset (comprising 20–25%) of breast carcinomas and being differently associated with hormone-related risk factors [46,47].

In this study we employed different MR approaches: IVW-based, Egger regression, and median-based methods. IVW methods are generally more powerful compared to the other methods implemented but they do not account for directional (unbalanced) pleiotropy. MR-Egger accounts for heterogeneity due to pleiotropy and detects directional pleiotropy. The median-based method is more robust to overall pleiotropy as it allows up to 50% of invalid instruments (i.e., violating the second or third assumptions of MR) [48].

Although we checked for associations between our genetic variants and confounders, a violation of the second assumption of the MR approach cannot be entirely excluded due to the potential presence of unknown confounding factors. Other limitations include that our study may be slightly biased due to uncontrolled confounding from family effects, such as assortative mating, dynastic effects, and population structure [49]. We minimized this bias from population stratification by performing a two-sample MR analysis in samples composed of Caucasian individuals only. However, this may cause our findings to not be generalizable to other ethnic groups. Referring to the body shapes, it is difficult to interpret the effect of PCs on breast cancer as it is not evident to what 1 SD corresponds with in terms of anthropometry. Finally, it is important to highlight that we were not able to differentiate between pre- and post-menopausal breast cancer with the available data.

## 5. Conclusions

This MR analysis provides evidence of an inverse association between a body shape that reflects general adiposity and the risk of overall breast cancer among women of European decent. The strength of association differed across sub-types of breast cancer suggesting a stronger inverse association for ER− sub-types as compared to ER+ sub-types. The most likely explanation for these inverse associations with overall breast cancer and sub-types is that the genetic instruments for this body shape indicate general adiposity during childhood/adolescence rather than during later adulthood. To test this assertion in future studies, genetic instruments for body shape during both childhood/adolescence and adulthood (age 30+ years) should be identified and their independent risk associations with breast cancer compared in a multivariable MR framework.

Furthermore, we found a weak positive association between a body shape characterizing tall women with a low WHR and overall breast cancer risk and most sub-types. This is in line with previous evidence on height and breast cancer. Our findings add to this evidence base by ruling out confounding by WHR.

A body shape characterizing tall women with a large WHR was not associated with breast cancer risk.

The current study provides complementary evidence to observational studies to interrogate multiple anthropometric measurements in relation to breast cancer risk.

## Figures and Tables

**Figure 1 cancers-15-01296-f001:**
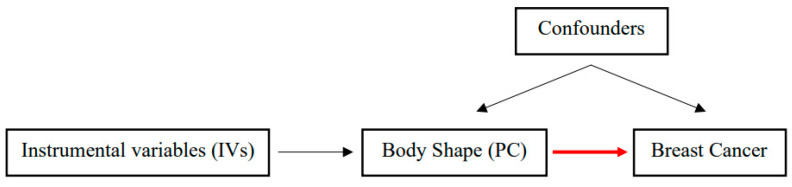
Causal directed acyclic graph (DAG) of Mendelian Randomization analysis. The red arrow highlights the causal association between the body shape (exposure) and breast cancer (outcome) assessed. PC, principal component.

**Figure 2 cancers-15-01296-f002:**
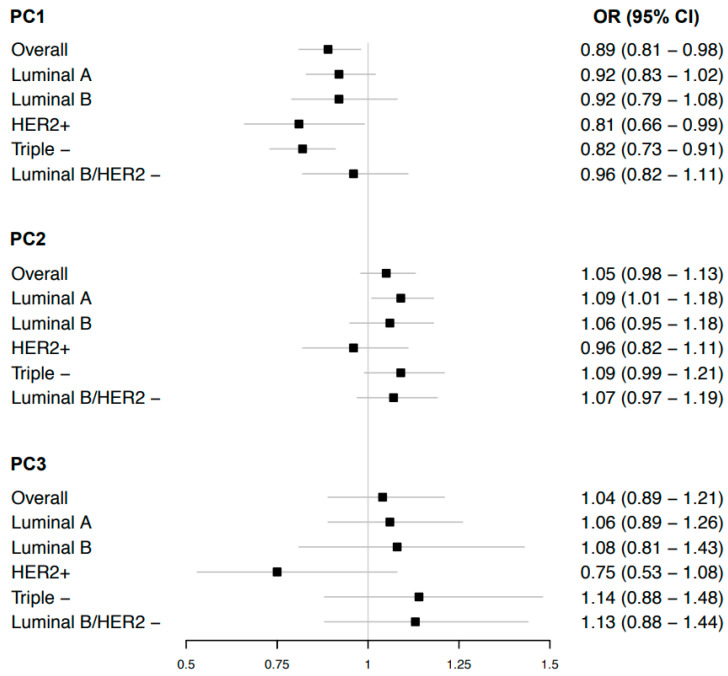
Mendelian randomization estimates between body shape phenotype PC1, PC2, and PC3 and overall breast cancer risk and sub-types. CI, confidence interval; OR, odds ratio; PC, principal component. We report random-effect IVW results due to heterogeneity across the genetic instruments.

## Data Availability

The data presented in this study regarding the body shape phenotypes are available in https://www.nature.com/articles/ncomms13357, accessed on 18 February 2021. Publicly available datasets were analyzed in this study regarding breast cancer risk. This data can be found here: https://bcac.ccge.medschl.cam.ac.uk/, accessed on 21 February 2021.

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
