# Peer review of "Body Shape Phenotypes and Breast Cancer Risk: A Mendelian Randomization Analysis"

_cancers, 2023, doi:10.3390/cancers15041296_

Round 1
Reviewer 1 Report
Authors performed two-sample Mendelian randomization analyses to assess the association between 188 genetic variants robustly linked to the first three principal components and BC. The reported that general adiposity was inversely associated with overall BC risk; tall with low WHR was weakly positively associated with overall BC risk, but with a confidence interval including the null; tall with large WHR was not associated with overall BC risk. They claimed the study provides evidence for potential causal associations between body shape and BC risk and BC sub-types highlighting the importance to also assess body morphology holistically. The conclusion is very interesting, but more evidence is required for population in such journal.
Obesity has been reported to be a risk factor for many cancers. In addition, as reported by many previous studies that weight gain is a major health risk for postmenopausal women. gaining weight after menopause increases their risk for chronic diseases including breast cancer. fats have been found to be associated with systemic inflammation and oxidative stress that can lead to metabolic syndrome, including insulin resistance, which can increase the risk of getting breast cancer. Accumulating abdominal adipose tissue does increase the risk of breast cancer in postmenopausal women (DOI: 10.1007/s10549-018-5016-3)(DOI:10.1210/clinem/dgac241)(DOI:10.1210/endocr/bqab195)(DOI:10.3390/nu14224926)(DOI:10.1093/jnci/82.4.286). However, this study concluded that general adiposity was inversely associated with overall BC risk, therefore bringing up a serious challenge to existed knowledge. Thus, evidence only from dry-lab seems weak and insufficient. In vitro or in vivo experiments are strongly suggested to added for supporting their conclusions.
Furthermore, the occurrence and development of breast cancer are closely related to endocrine level and immune environment. Therefore, a section should be added to particularly discuss the potential mechanisms of the observed phenomenon of this study. The characteristics of the studied population should also be provided and described in manuscript (such as age, menopause).
Reviewer 2 Report
The paper is very interesting and in my opinion, has a potential for application.
On the other hand, the parameters tested in the current study do not necessarily indicate a chance of getting breast cancer.
Mammography examination A test for early diagnosis using a mammography device, is a quick x-ray examination of the breasts. A mammogram scan test is intended for women without any symptoms, and its purpose is to detect a cancerous growth in the breast in its early stages, when initial malignant changes develop in the breast, such as small lumps or calcifications, which cannot be felt. This is a simple process that takes a few minutes.
In my opinion, this is the best indicator of breast cancer as well as a genetic examination.
The authors should revise the paper according to the following comments:
- The introduction section is too short. Breast cancer has been studied a lot by different researchers from all fields around the world. Please keep up to date on research from recent years in the field of breast cancer research.
- The method section is too short. After that, the authors present the results of the paper. It is too early to present the results after a few lines.
- The following paper can be added to the current article:
- Noray, L. P., Dimou, N., Sedlmeier, A. M., Fervers, B., Romieu, I., Viallon, V., Gunter, M. J., Ferrari, P., Carreras-Torres, R., & Freisling, H. (2022). P46 Body shape phenotypes and breast cancer risk: a Mendelian randomization analysis. In SSM Annual Scientific Meeting. Society for Social Medicine Annual Scientific Meeting Abstracts. BMJ Publishing Group Ltd. https://doi.org/10.1136/jech-2022-ssmabstracts.140
- Nave, Op., & Sigron, M. (2022). A Mathematical Model for the Treatment of Melanoma with the BRAF/MEK Inhibitor and Anti-PD-1. In Applied Sciences (Vol. 12, Issue 23, p. 12474). MDPI AG. https://doi.org/10.3390/app122312474
- Sedlmeier, A. M., Viallon, V., Ferrari, P., Peruchet-Noray, L., Fontvieille, E., Amadou, A., Seyed Khoei, N., Weber, A., Baurecht, H., Heath, A. K., Tsilidis, K., Kaaks, R., Katzke, V., Inan-Eroglu, E., Schulze, M. B., Overvad, K., Bonet, C., Ubago-Guisado, E., Chirlaque, M.-D., … Freisling, H. (2022). Body shape phenotypes of multiple anthropometric traits and cancer risk: a multi-national cohort study. In British Journal of Cancer. Springer Science and Business Media LLC. https://doi.org/10.1038/s41416-022-02071-3
- The discussion section is a very poor presentation. Please extend this section.
- The conclusion section is very poor. Please extend extensively this section.
In conclusion. My first impression from reading the article is that the authors rushed to submit the article to the journal. The article should be completely rewritten. present it in a more professional manner. with more detailed explanations. Explain why the current article and research is superior to other research done in this exact area.
Round 2
Reviewer 1 Report
Authors summarized a relevant in vivo study, which showed in an animal model that tissue derived estrogen in overweight adolescent rats (immediately after puberty) may induce early breast differentiation and render the breast tissue less susceptible to tumorigenesis, which is a strong evidence to support their conclusion. More relevant evidence (in vivo or in vivo) from existing studies is also required, and should be summarized as a single section in manuscript.
Reviewer 2 Report
The authors revised the paper. Please take care again and revised the paper according to all of the comments. In addition, the paper should edit by English mother language.
Round 3
Reviewer 2 Report
Please revise the paper according to the comment as attached in the previous file.